

# Precipitation regime and stable isotopes at Dome Fuji, East Antarctica

A. Dittmann[1], E. Schlosser[1,2], V. Masson-Delmotte[3], J. G. Powers[4], K. W. Manning[4], M. Werner[5], and K. Fujita[6]

[1]Inst. of Atmospheric and Cryospheric Sciences, University of Innsbruck, Innsbruck, Austria
[2]Austrian Polar Research Institute, Vienna, Austria
[3]Laboratoire des Sciences du Climat et de l'Environnement, Gif-sur-Yvette, France
[4]National Center for Atmospheric Research, Boulder, CO, USA
[5]Alfred Wegener Institute, Bremerhaven, Germany
[6]Graduate School of Environmental Studies, Nagoya University, Chikusa-ku, Nagoya, Japan

*Correspondence to:* Anna Dittmann (Anna.Dittmann@student.uibk.ac.at)

**Abstract.** A unique set of one-year precipitation and stable water isotope measurements from the Japanese Antarctic station Dome Fuji has been used to study the impact of the synoptic situation and the precipitation origin on the isotopic composition of precipitation on the Antarctic Plateau. The Antarctic Mesoscale Prediction System (AMPS) archive data are used to analyse the synoptic situations that cause precipitation. These situations are investigated and divided into five categories. The most

common weather situation during a precipitation event is an upper-level ridge that extends onto the Antarctic Plateau and causes strong northerly advection from the ocean. Most precipitation events are associated with an increase in temperature and wind speed and a local maximum of $\delta^{18}O$. During the measurement period, 21 synoptically-caused precipitation events caused 60 % of the total annual precipitation, whereas the remaining 40 % were predominantly attributed to diamond dust. By combining the synoptic analyses with 5-day back-trajectories, the moisture source regions for precipitation events were

estimated. An average source region around a latitude of 55 °S was found. The atmospheric conditions in the source region were used as initial conditions for running a Rayleigh-type isotopic model in order to reproduce the measured isotopic composition of fresh snow and to investigate the influence of the precipitation source region on the isotope ratios. The model represents the measured annual cycle of $\delta^{18}O$ and the second-order isotopic parameter deuterium excess reasonably well, but yields, on average, too little fractionation along the transport/cooling path. While simulations with an isotopic general circulation model

(GCM) (ECHAM5-wiso) for Dome Fuji are on average closer to the observations, this model cannot reproduce the annual cycle of deuterium excess. In the event-based analysis, no evidence for a correlation of the measured deuterium excess with the latitude of the moisture source region or the corresponding conditions there was identified. Contrary to the assumption used for decades in ice core studies, a more northern moisture source does not necessarily mean a larger temperature difference between source area and deposition site and thus precipitation that is more depleted in heavy isotopes and has a higher deuterium excess.



# 1 Introduction

Ice cores from the Antarctic ice sheet have been used successfully for several decades in paleoclimatology to provide high resolution records of past changes in snow and ice isotopic composition, a key tool to reconstruct past local temperature changes (Masson-Delmotte et al., 2006). A linear relationship between the annual mean air temperature and the stable water isotope ratios (D/H, $^{18}O/^{16}O$) of the snow/ice at the drilling site has been found (Dansgaard, 1964; Lorius et al., 1969). The ratio of stable water isotopes is usually given with respect to the Vienna Standard Mean Ocean Water (V-SMOW) in the $\delta$-notation:

$$\delta^{18}O = \left[ \frac{\left( \frac{^{18}O}{^{16}O} \right)_{sample}}{\left( \frac{^{18}O}{^{16}O} \right)_{SMOW}} - 1 \right] \text{‰}$$

However, Dansgaard (1964) has developed a theoretical framework showing that the isotopic composition of precipitation does not only depend on condensation temperature ($T_c$), but also on the distillation occurring along the air mass trajectory, from the first evaporation from an oceanic source to the final deposition at the ice core drilling site. In the light of future climate changes, it is essential to fully understand past temperature fluctuations (Masson-Delmotte et al., 2006). The source region of precipitation plays an important role, as it determines the meteorological conditions during the first evaporation and the length and position of the moisture transport path (Dansgaard, 1964; Schlosser et al., 2004), the latter being determined by the large-scale synoptic situation. Schlosser et al. (2004) found distinct differences in isotope ratios of fresh snow samples at Neumayer Station, East Antarctica, depending on the moisture origin. Masson-Delmotte et al. (2008) found a strong spatial variation of the local isotope-temperature slope. They suggested that one reason for this is the spatial variability in moisture origin.

Quantitative temperature reconstructions from ice core records have been developed using combined water stable isotopes (D and $^{18}O$) in order to assess from the second order isotopic parameter deuterium excess past changes in moisture source characteristics (e.g. sea surface temperature (SST), relative humidity (RH)) and improve the Antarctic temperature estimate (Stenni et al., 2001; Uemura et al., 2012). However, very few studies have been performed on present day Antarctic precipitation and the relationships between its isotopic composition and moisture source and atmospheric transport characteristics. For a better understanding of the proxy data from ice cores, it is necessary to analyse the processes determining the isotopic composition of the snow at the drilling locations for recent time periods, for which both meteorological and stable isotope data are available.

The origin of precipitation is determined by the synoptic weather situations producing snowfall. Snowfall associated with synoptic precipitation events is mostly associated with warm events, which may be recorded in water stable isotopes Noone et al. (1999). While in the coastal areas precipitation is caused by frontal activity in the circumpolar trough, different precipitation mechanisms are at play on the high Antarctic plateau. Generally, two main types of precipitation can be distinguished here. The first is that of diamond dust or clear-sky precipitation, which consists of very fine ice needles and is formed by radiative cooling of a nearly-saturated air mass. The second is one in which distinctly higher amounts of snowfall occur due to amplified Rossby waves that lead to advection of relatively warm and moist air to the interior of the continent. This occurs several times per year. When moist oceanic air is flowing southward, it is lifted by the steep slopes of the Antarctic continent, thereby adia-





batically cooled. When saturation is reached, precipitation starts to fall and the absolute humidity of the air decreases rapidly towards the plateau. It has been estimated that on the high Antarctic Plateau approximately 25 - 60 % of the annual precipitation originates from a few synoptic precipitation events (Reijmer and van den Broeke, 2003; Schlosser et al., 2010a; Ekaykin et al., 2004). The remaining precipitation results from diamond dust. The spatial distribution of the precipitation is highly influenced by the topography (King and Turner, 1997; Schlosser et al., 2008a).

The present study aims at an improved understanding of the atmospheric influences on the isotopic composition of Antarctic precipitation. Of particular interest is the influence of the position of the moisture source region and the conditions in this area and along the moisture transport paths.

To reach this goal, precipitation conditions at the East Antarctic deep ice core drilling site Dome Fuji were investigated using the unique data set of daily precipitation and stable isotope measurements by Fujita and Abe (2006). The atmospheric flow conditions and synoptic situations causing precipitation events at Dome Fuji were analysed with help of Antarctic Mesoscale Prediction System (AMPS) archive data. Moisture source areas for the precipitation events were estimated with the combined information of backwards trajectory calculations and synoptic weather charts.

Furthermore, the ability of a simple Rayleigh-type isotopic model and an isotopically-enhanced general circulation model (GCM) to reproduce the measured isotopic composition of precipitation was investigated. The conditions at the estimated source regions were used as initial conditions for the Rayleigh-type isotope model. Finally, the influence of the precipitation origin and the general atmospheric flow on the stable isotopes and, in particular, on the deuterium excess, were investigated using the combined information from synoptic analysis, trajectory calculations, observational data, and isotopic modelling.

## 2 Previous work

### 2.1 Synoptic analyses

In the past, several studies were conducted to investigate the synoptic situations causing high-precipitation events on the plateau of Dronning Maud Land (DML) (Sinclair, 1981; Noone et al., 1999; Birnbaum et al., 2006; Schlosser et al., 2010a; Hirasawa et al., 2013). One of the first to study the intrusion of synoptic weather systems into the interior of Antarctica was Sinclair (1981). He found that in December 1978 an intensifying ridge over DML and cyclogenesis in the Weddell Sea caused record-high temperatures in the interior of Antarctica. Noone et al. (1999) used reanalysis data from ECMWF to analyse high-precipitation events in DML. They found that the most common weather situation causing high-precipitation events develops through amplified long waves with a cyclone in the Weddell Sea and an Atlantic block forcing enhanced northerly advection. Schlosser et al. (2010a) investigated synoptic situations causing high precipitation rates at Kohnen using AMPS archive data. Kohnen lies northwest of Dome Fuji and is around 1000 m lower in altitude (Figure 1). Therefore, it is more influenced by the coastal climate than Dome Fuji. Schlosser et al. (2010a) distinguished between five weather situations that cause high precipitation events: a strong deep cyclone over/north of Kohnen, a blocking high east of Kohnen Station with northwesterly flow, a blocking high east of Kohnen Station with northeasterly flow, a (weak) ridge above Kohnen Station and an upper-air low south of Kohnen.





At Dome Fuji, no investigation about the frequency of different weather situations has been carried out so far. However, in several case studies, blocking events that caused large amounts of precipitation were analysed (Enomoto et al., 1998; Hirasawa et al., 2000; Schlosser et al., 2010b; Hirasawa et al., 2013). The investigated precipitation events were associated with a strong increase of wind speed, a breakdown of the surface inversion and a large temperature increase.

## 2.2 Origin of precipitation

Since the origin of precipitation largely influences its stable isotope ratios, it has been studied by various authors. The first studies combined surface snow isotopic data and theoretical Rayleigh distillation calculations, and concluded that the origin of inland Antarctic precipitation was mainly situated in the subtropics (Jouzel and Merlivat, 1984; Petit et al., 1991; Motoyama et al., 2005). However, more recent studies with trajectory models have found source regions much further south (Reijmer et al., 2002; Helsen et al., 2006). Reijmer et al. (2002) determined an average source region between $50\,°$ and $60\,°$S for precipitation at Antarctic ice core drilling sites. Suzuki et al. (2008) used a trajectory model and ERA-40 reanalysis data to investigate the transport towards Dome Fuji for the year 1997. They distinguished between clear and snowy conditions using observations of snowfall and clouds. The backward trajectories were calculated for five days and arrived at $500\,hPa$. They found that during snowy conditions in winter the travel time of 5 days was sufficient to identify the evaporation area at the sea surface. In summer, the trajectories were often positioned entirely above the continent and at greater height, thus not yielding information about moisture origin. However, the authors did not distinguish between clear-sky precipitation and synoptic precipitation, and the cloud observations had large uncertainties. Sodemann and Stohl (2009) studied the seasonality of moisture sources for Antarctic precipitation using Lagrangian moisture source diagnostic. They traced water vapour transport for 20 days backwards in time to estimate the precipitation origin for Antarctica and found a source region for Dome Fuji at a mean latitude of $44\,°$S. They state that their results are consistent with findings from GCM with tagged tracers. However, it is not trivial to understand the dynamics of the calculated transport. The shorter trajectories in the present study were cross-checked with the synoptic analysis.

## 2.3 Stable isotopes

Commonly, the second-order isotopic parameter deuterium excess ($d = \delta D - 8 \cdot \delta^{18}O$) is used to extract information about the moisture source region for precipitation in ice cores (Stenni et al., 2001; Masson-Delmotte et al., 2004; Uemura et al., 2012). The deuterium excess reflects the different behaviour of $HD^{16}O$ and $H_2{}^{18}O$ during fractionation. On average, the impact of the equilibrium fractionation on global precipitation is 8 times higher for $HD^{16}O$ than for $H_2{}^{18}O$. The kinetic fractionation is nearly equal for both isotopes. The deuterium excess is thus a way to quantify the kinetic fractionation (Jouzel, 2014). Kinetic fractionation during evaporation is dependent on the meteorological conditions at the time: SST, RH, and wind speed (Jouzel and Merlivat, 1984). Uemura et al. (2008) measured the isotopic composition of moist air at $15\,m$ altitude in the Southern Ocean from $30\,°$ to $65\,°$S. They found an anti-correlation of d with RH at the measurement height and a positive correlation with SST. They showed that RH explains variations of d on short timescales. Steen-Larsen et al. (2014) recently performed in





situ measurements of d in the marine boundary layer of the north Atlantic. They could explain 84 % of the d variance by RH. This relationship was dependent on SST. However, they could not find any correlation of d with wind speed.

In paleoclimatology, the deuterium excess from ice cores was used to extract information about the conditions at the moisture source region (Stenni et al., 2001; Uemura et al., 2012). Note that without additional information one cannot distinguish
between a change of the meteorological conditions at a given moisture source and a change in the position of the moisture source. It is also assumed that the signal of the initial deuterium excess is preserved in final precipitation. However, during the moisture transport, additional kinetic fractionation occurs in the clouds, mainly during deposition on ice crystals. In mixed clouds the Bergeron-Findeisen effect induces kinetic fractionation. It is caused by the difference in the saturation vapour pressure over water and ice which leads to a net evaporation of liquid droplets and a net deposition of vapour on ice crystals
(Ciais and Jouzel, 1994). Apart from non-equilibrium processes, d is also influenced by the final temperature during deposition: Since the equilibrium fractionation coefficients are temperature dependent, the slope between $\delta D$ and $\delta^{18}O$ is smaller than 8 for low temperatures. This leads to an increase of d with decreasing temperature. Also the role of kinetic fractionation at supersaturation on ice crystals could increase the anti-correlation of d with temperature (Masson-Delmotte et al., 2008). The relative importance of the processes determining d in Antarctic snow is not understood sufficiently.

To reconstruct information about the temperature both at the deposition site and at the source region, simple isotopic models are used. A frequently-used model is the mixed cloud isotope model (MCIM) (Ciais and Jouzel, 1994). It calculates the isotopic fractionation of an isolated air parcel on the cooling path from the first evaporation to the final deposition. It considers equilibrium and kinetic fractionation processes. Several studies have tried to test the performance of the model to reproduce the present day isotopic composition of snow from measurements (Jouzel and Merlivat, 1984; Petit et al., 1991; Uemura et al.,
2012). To include atmospheric dynamics and to find appropriate source conditions some studies have combined Rayleigh-type models with trajectory models. For example, Schlosser et al. (2008b) used MCIM and a trajectory model to investigate the relationship between the seasonal cycle of deuterium excess and the precipitation origin with snow samples from Neumayer station. The model could reproduce the annual cycle of $\delta^{18}O$ and $\delta D$ but the amplitude was underestimated. They found on average a lower d for trajectories originating from lower latitudes than for trajectories originating from the Antarctic continent.
Helsen et al. (2006) combined a five-day backward trajectory model and MCIM, to investigate isotope records from snow pits from four locations in western DML. Instead of source conditions at the sea surface, they used the monthly mean vapour isotopic composition from ECHAM4 at the start height of the trajectories above the ocean. The model was able to simulate the seasonal cycle of the measured values, but at the low temperatures in the Antarctic interior, the isotopic depletion was underestimated.

# 3   Study site

The Japanese ice core drilling station Dome Fuji (Dome F) (77.31 °S, 39.70 °E) is situated on the East Antarctic plateau in Dronning Maud Land (DML) at an elevation of 3810 m. The distance to the coast is approximately 1000 km. The continental location at high southern latitudes and the high elevation of the station are the reasons for its extremely cold and dry climate,



with a mean annual surface mass balance of $27.3\,\mathrm{mm}$ w.e. (Motoyama et al., 2005) and a mean annual temperature of $-57.7\,^{\circ}\mathrm{C}$ (Kameda et al., 2008). Dome F is one of the places with the oldest ice in Antarctica (Motoyama, 2007). Two ice cores have been drilled, the first one in 1996 with a maximum age of $340{,}000\,\mathrm{yr}$ (Watanabe et al., 1999). The second core was completed in 2007 at a depth of $3035.2\,\mathrm{m}$, which covered the past 720,000 years (Motoyama, 2007). This is the second-oldest ice ever recovered, exceeded only by the EPICA Dome C core (EPICA Community Members, 2004).

## 4 Data

### 4.1 Precipitation and stable isotopes

Fujita and Abe (2006) were the first to perform direct precipitation measurements and sampling for isotopic measurements on the Antarctic Plateau. Daily snow samples were collected in plastic containers, placed on the station roof. The observation height of $4\,\mathrm{m}$ minimized the measurement error due to drifting snow. The precipitation amount and the stable isotope ratios of the samples were measured from 3 February 2003 to 20 January 2004. The samples were analysed in Japan with a mass spectrometer by the $CO_2 - H_2O/H_2 - H_2O$ equilibrium method. The accuracy of the measurements of $\delta^{18}O$ is $0.05\,\text{\textperthousand}$ and of $\delta D$ $0.5\,\text{\textperthousand}$. The precision of d is $0.64\,\text{\textperthousand}$. The advantage of direct precipitation measurements compared to accumulation measurements is that alterations through wind scouring and sublimation after the snowfall do not have to be considered. Similarly, the stable isotope ratios of the precipitation samples are not affected by post-depositional processes, such as exchange with the atmosphere at the snow-air interface or diffusion within the snowpack/ice.

### 4.2 AWS data

The Antarctic Meteorological Research Center (AMRC) and Automatic Weather Station (AWS) Program are sister projects of the University of Wisconsin-Madison funded under the United States Antarctic Program (USAP). These projects focus on data for Antarctic research support, providing real-time and archived weather observations and satellite measurements and supporting a network of automatic weather stations across Antarctica. The current AWS at Dome Fuji, which measures the standard meteorological variables of air temperature, pressure, wind speed, wind direction, and humidity, was set up by the AMRC in February 1997. Data can be obtained from the AMRC website (http://amrc.ssec.wisc.edu).

### 4.3 AMPS archive data

For the analysis of the weather situations and trajectory calculation, the Antarctic Mesoscale Prediction System (AMPS) archive data (Powers et al., 2003) was used. AMPS has been created for providing numerical weather prediction guidance for the forecasters of the USAP, but has also widely been used for scientific studies (Schlosser et al., 2010a, 2008a; Uotila et al., 2009) For the 2003 – 2004 period analysed here, AMPS employed Polar MM5, a mesoscale atmospheric model run at high resolution and adapted to the special conditions of the polar environment. It is a version of the fifth-generation Pennsylvania State University-NCAR Mesoscale Model, with modifications made to better represent conditions in polar regions (Powers et al.,





2003). These include an improved representation of sea ice which allowed fractional sea ice coverage in grid cells and using the latent heat of sublimation for calculating heat fluxes over ice surfaces. Other modified areas were revised cloud/radiation interactions and ice phase microphysics, and an improved treatment of heat transfer through snow and ice surfaces (Bromwich et al., 2001). Schlosser et al. (2008a) compared AMPS precipitation in western DML to the surface mass balance derived from glaciological data for the years 2001–2006. They found similar spatial patterns in both data sets, related to topography and prevailing wind systems. However, they noted that on the high Antarctic Plateau precipitation might be underestimated by the model because the MM5 does not include the formation of diamond dust.

AMPS was run with a set of nested domains of varying extents and grid sizes. In this study, the gridded model output used was that from the outermost domain with a gird spacing of 90 km and its nest, with a grid of 30 km.

# 5 Methods

## 5.1 Definition of precipitation events

To quantify the frequency of precipitation events and to compare measurements and model results, a threshold was defined to distinguish precipitation events from diamond dust. In most cases, the amount of precipitation is considerably higher for dynamically-caused events than for diamond dust events. However, sometimes it is difficult to distinguish between a small amount of synoptic precipitation and diamond dust (Schlosser et al., 2010a). Furthermore, the errors in both measured and modelled precipitation for specific events can be large. In previous studies, various ways have been found to determine a threshold for precipitation events, e.g. 2 mm w.e. at 75 °S and 0 °E for precipitation events in DML (Noone et al., 1999), a threshold with which 50 % of the total precipitation is counted as snowfall (Reijmer et al., 2002) or a deviation from the mean of one standard deviation (Fujita and Abe, 2006). A reasonable way to define precipitation events is to consider percentiles (Schlosser et al., 2010a). This accounts for the fact that precipitation does not have a Gaussian distribution. Here, from the frequency distribution, the 90 % percentile was chosen as the threshold. For this definition it was necessary that all precipitation events for which a synoptic origin was visible on the weather charts and which were accompanied by increased temperature and wind speeds exceeded the threshold.

## 5.2 Trajectory calculation

To estimate the source region of precipitation events, three-dimensional backwards trajectories were calculated using the graphics software RIP ("Read/Interpolate/Plot") (Stoelinga, 2009) and AMPS archive data. The three-dimensional displacement of an air parcel is calculated using the following iterative scheme and an iteration step $\Delta t$ of 30 min:

$$X_{n+1} = X_0 + \frac{\Delta t}{2}[v(X_0, t) + v(X_n, t + \Delta t)]$$





$X_{n+1}$ is the $(n+1)^{th}$ iterative approximation of the position vector at the time $t + \Delta t$. It is calculated from the position vector of the air parcel at time t ($X_0$), the wind vector at the position $X_0$ and time t ($v(X_0, t)$) and the wind vector at the position of the previous iteration and time $t + \Delta t$ ($v(X_n, t + \Delta t)$).

### 5.3 Isotope Modelling

Generally, two types of isotopic models are distinguished: Rayleigh-type models (e.g. Merlivat and Jouzel, 1979) consider the fractionation in an isolated air parcel using only moisture source and condensation conditions as input. They do not include dynamic processes or turbulent mixing. Isotopic GCMs (e.g. ECHAM5-wiso, Werner et al. (2011)) include an explicit representation of stable water isotopes into a three dimensional atmospheric model. Equilibrium and kinetic fractionation processes are calculated for each phase change (Jouzel, 2014).

Ciais and Jouzel (1994) included kinetic fractionation processes in mixed clouds into a Rayleigh-type model developed by Merlivat and Jouzel (1979), resulting in the so-called Mixed Cloud Isotope Model (MCIM). In contrast to the original Rayleigh-type model, an adjustable fraction of the condensate is kept in the cloud. In an also adjustable range of temperatures, the cloud can hold both liquid droplets and ice crystals. In this temperature range, additional kinetic fractionation processes occur due to the Bergeron-Findeisen process. The initial isotopic composition after evaporation can be derived either from the assumption that evaporation and precipitation are balanced on a global scale (closure equation), or it can be taken from an isotopic GCM at the required height. Jouzel and Koster (1996) found a systematic bias when using the closure equation on a regional scale. But the advantage of the closure equation over the GCM approach is that the relationship between the source conditions and the calculated or observed isotopes can be investigated directly. Therefore, this approach was chosen here. As input, the closure equation uses SST, the mean sea level pressure, RH calculated at SST, and wind speed at the evaporation area. The maximum wind speed allowed in MCIM is $10\,\mathrm{ms}^{-1}$. In $30\,\%$ of the cases the wind speed in the estimated source regions was higher and had to be adjusted. The initial conditions for the closure equation were taken from ECMWF Interim Reanalysis (ERA-Interim) at the estimated source regions of the precipitation events. The required variables are available from ECMWF (http://apps.ecmwf.int/datasets/data/interim-full-daily/). The arrival conditions were taken from AMPS at the arrival height of the trajectories.

Furthermore, simulations with the isotopic GCM ECHAM5-wiso (Werner et al., 2011) were used. This model enhances the atmospheric GCM ECHAM5 by stable water isotope diagnostics. A similar approach already was performed with the predecessors ECHAM3 (Hoffmann et al., 1998) and ECHAM4 (Werner et al., 2001). Into the water cycle of ECHAM5 additional to $H_2O^{16}$ water containing $O^{18}$ and D is implemented. When a phase changes of a water mass takes place, fractionation processes are considered. Werner et al. (2011) compared the output of ECHAM5-wiso with varying horizontal and vertical resolution to observational data. They found a good agreement of the simulations with observational data on a global scale. A higher horizontal and vertical resolution clearly improved the results. For the Antarctic continent, they detected a warm bias of the surface temperatures from ECHAM5. Therefore, the model precipitation was less depleted of heavy isotopes than the observations there. In this study, an ECHAM5-wiso simulation with a grid size of $1.125\,^\circ \times 1.1215\,^\circ$ and 31 vertical levels for the point of Dome F were used. The simulation was performed for the period 1979 to 2013, with prescribed values of ocean





surface conditions (SST, sea ice concentration), insolation and atmospheric greenhouse gas concentrations. Monthly SST and sea ice concentration were derived from the ERA-interim data set, and the dynamic-thermodynamic state of the atmosphere was nudged to ERA-interim data, as well.

In this study, we will compare the Dome F precipitation isotopic measurements with two different types of model outputs. First, we will use atmospheric backtrajectories combined with MCIM, a theoretical distillation model, with hypotheses of a single moisture source. Second, we will use the outputs of ECHAM5-wiso, a GCM equipped with water stable isotopes and nudged to atmospheric reanalyses. The advantage of the first approach is to make best use of regional atmospheric circulation data, albeit with simplifications for isotopic processes. The advantage of the second approach is the physically consistent framework of the global atmospheric model, albeit with the usual caveats of model resolution and physical biases.

## 6 Results

### 6.1 Precipitation and temperature

The total precipitation in the measurement period from 3 February 2003 to 20 January 2004 was 27.5 mm w.e.. The mean daily precipitation was 0.08 mm w.e.. On only 24 days no precipitation was observed. Figure 2a shows the time series of the direct precipitation measurements. It shows several high precipitation events on single days or on a few consecutive days and only low amounts of diamond dust during the rest of the time. The highest observed daily precipitation is 2.1 mm w.e. on 15 December 2003.

In Fig. 2a the daily AMPS precipitation is shown by the red dots. The total modelled precipitation in the measurement period is 16.0 mm, considerably less than the measured precipitation. For many days with large measured precipitation amounts, AMPS forecasts increased precipitation on the same day or shifted by one day. However, the predicted precipitation amount is mostly too low compared to the observations.

The mean daily temperature ranges between -77.8 °C and -25.3 °C. The average for the entire period is -54.7 °C. The measurements show a high variability (Fig. 2b). Especially in winter and spring time, several short-term temperature increases of more than 20 °C were observed. These rapid temperature rises are mostly associated with an increased amount of precipitation.

Figure 3 displays the frequency distribution of the measured and modelled precipitation. On 194 out of 337 days, precipitation amounts less than 0.03 mm w.e. were observed. The frequency of the values decreases rapidly with increasing precipitation. The histogram of the model precipitation differs from that of the measurements mainly in a higher frequency of values between 0 and 0.015 mm and in no model values for precipitation amounts over 0.6 mm. The 90 % percentile, which is the threshold for precipitation events, is marked in Fig. 2a and Fig. 3. It has a value of 0.16 mm w.e. for the measurements and 0.12 mm for AMPS precipitation. This leads to 34 days with precipitation events from observations, and 31 days from AMPS. Taking into account that a precipitation event can consist of several days yields 21 precipitation events for the measurements and 19 for the AMPS simulations. With this threshold, 60 %, of the measured and 43 % of the modelled amount of precipitation is caused by precipitation events. 57 % of the 21 measured precipitation events can be assigned to a modelled precipitation event.





## 6.2 Stable isotopes

Figure 2c shows the time series of the daily measured $\delta^{18}O$ in precipitation. As expected from isotopic distillation, the course of $\delta^{18}O$ mostly follows the temperature development throughout the year. The mean $\delta^{18}O$ is -61.3‰ with a minimum of -81.9‰ in August and a maximum of -33.0‰ in December. The annual cycle of d (Fig. 2d) is anti-correlated to $\delta^{18}O$ and

temperature with a maximum in winter and low values in summer. The deuterium excess varies between -52.6‰ and 66.9‰ and has a mean value of 17.4‰ in the measurement period.

In Fig. 2, the days with an observed precipitation event are marked by black circles. Additionally, to distinguish between larger and smaller precipitation events, days where the precipitation is higher than the 95 % percentile are marked with black dots. $\delta^{18}O$ often has a local maximum during precipitation events. It mostly corresponds to a local temperature maximum

at the same time. On average, $\delta^{18}O$ is 4‰ higher on days with a precipitation event than for days with diamond dust. The behaviour of deuterium excess is only uniform in winter where from June to August mostly a local minimum is observed during precipitation events.

Figure 4a displays the scatter plot of $\delta^{18}O$ versus temperature. The green crosses represent all daily measurements throughout the measurement period. The dark green line is the associated linear fit with $\delta^{18}O = 0.76 \pm 0.02 \cdot T - 19.1 \pm 1.3$ as regression

equation and a correlation coefficient R of 0.88. The $\delta^{18}O$–T slope is close to the average Rayleigh distillation slope of 0.8. The red crosses show the measurements from days with precipitation events. The correlation coefficient of 0.93 for these days is higher than for the whole data set. The corresponding linear equation is $\delta^{18}O = 0.69 \pm 0.05 \cdot T - 23.2 \pm 2.6$, plotted in Fig. 4a as a dark red line. It has a slightly lower slope than the regression for the whole period. The relationship between deuterium excess and $\delta^{18}O$ is plotted in Fig. 4b. The anti-correlation of the two parameters is clearly visible. The slope between

20    d and $\delta^{18}O$ increases for lower $\delta^{18}O$ values, which correspond to lower temperatures. For precipitation events, the range of deuterium excess is smaller than for all measurements.

## 6.3 Analysis of synoptic situations causing precipitation events

The synoptic situations of the precipitation events were divided into five categories using a manual classification scheme based on the 500 hPa geopotential height fields from the AMPS archive. For each category, one example is shown.

### 6.3.1 Amplified High Pressure Ridge

In this situation, Dome Fuji is situated underneath an extended high pressure ridge with strongly amplified planetary waves northwest of Dome F. An example occurred from 1 to 2 August 2003. Figure 5a shows the 500 hPa geopotential height for 1 August 12 UTC. A trough in the Atlantic Ocean with large meridional extent leads to strong advection from the southern mid-latitudes (north of 50 °S) towards DML. East of the trough, an extended ridge expands towards the south over the entire

western half of East Antarctica. The strong northerly flow leads to the advection of relatively warm and moist oceanic air masses onto the Antarctic Plateau. During the ascent of the air onto the plateau, it is cooled adiabatically, and orographic precipitation forms. The largest precipitation amounts are found north of Dome Fuji on the slope of the plateau, as can be





seen in Fig. 5b. When the air-mass reaches the high plateau, most of the moisture has already fallen out and the amount of precipitation considerably decreases towards Dome F.

### 6.3.2 Weak High Pressure Ridge

The second situation is similar to the previous case, but with less strongly amplified waves. A weak high pressure ridge stretches over Dome F. The moisture origin of the precipitation thus lies south of $50\,°$S. On 14 August 2003, an event like this occurred. The $500\,\mathrm{hPa}$ geopotential height for 13 August shows - as in the previous case - a ridge progressing southward on the Antarctic Plateau and causing northerly advection from the coast towards the interior of the continent (Fig. 6a). But in contrast to the previous case, the waves are not as strongly amplified. The source region of the precipitation in this case is the Southern Ocean.

### 6.3.3 Blocking High

In this scenario, a stable high pressure system remains for several days above Dome F. It thus induces stronger advection of relatively warm and moist air than in the situations described before. An intense blocking situation occurred from 31 October to 6 November 2003. Figure 7a shows the situation on 3 November, when the most precipitation was measured. A stationary ridge stretches over large parts of East Antarctica towards the South Pole. Between the ridge and the trough west of it a strong northerly advection from the mid-latitudes takes place. In the following days, an upper-level high is cut off and stays at this position until 6 November. The advance of the ridge is accompanied by a strong warm air intrusion onto the continent. From 30 October to 2 November, the upper-air temperature increases by $18\,°$C above Dome Fuji. The warm air mass has a relatively high absolute humidity. This causes high precipitation on 3 August on the slope of the Antarctic Plateau that reaches also Dome F (Fig. 7b).

### 6.3.4 Southerly Flow

In this situation, Dome Fuji is situated between an upper-level low to the east and a ridge to the west. West of Dome Fuji, the air is advected from the north, then moves anticyclonically around the ridge, thus reaching Dome Fuji from a southerly direction. Because of the long transport path over the continent, this weather situation does not always cause precipitation at Dome F. Although it is a rather infrequent weather condition, it is counted as an own category because of its distinctive characteristics. An example occurs on 1 October 2003 (Fig. 8a). In this case, the forecast and the observed precipitation amount do not match well. AMPS simulates precipitation only in the western part of the continent where the terrain rises slowly to the plateau (Fig. 8b). The measurements, however, show a precipitation event of $0.24\,\mathrm{mm\,w.e.}$ on 1 October. Possibly the amount of transported moisture was underestimated by the model.

### 6.3.5 Previous Precipitation Event

The last type of precipitation event differs from the other categories in that the synoptic situation responsible for transporting the moisture onto the Antarctic Plateau has changed by the day of the high precipitation. Relatively moist and warm air from





an event that took place on a previous day in the vicinity is advected towards Dome F. When the air starts to cool over the continent, new precipitation forms. At the time when precipitation is observed at Dome F, no distinct synoptic situation causing it is visible any more. But often moisture and temperature are still elevated. If the lower layer of the air contains a high amount of moisture and is cooled to the dewpoint, also hoar frost can develop that is then counted as precipitation. Large amounts

of precipitation can result from these mechanisms. AMPS often underestimated the precipitation amount during this kind of event.

The event of 14 – 16 December 2003 provides an example. On 14 December, a cut-off high is situated over Dome F (Fig. 9a). It was cut off from a ridge stretching from the north-east onto the Antarctic Plateau. Dome F is positioned west of the centre of the cut-off high in a north-easterly flow. Extremely high snowfall was observed with $2.1\,\mathrm{mm\,w.e.}$ on 15 December

and $1.4\,\mathrm{mm\,w.e.}$ on 16 December – the highest values observed in the whole measurement period. AMPS simulates patches of precipitation around Dome F (Fig. 9b). The precipitation is probably caused by local orographic lifting and radiative cooling.

### 6.3.6 Frequency distribution of the weather situations

The frequency distribution of the described synoptic patterns for days with a measured precipitation event is shown in Fig. 10a. Note that due to the brevity of the measuring period, our results have no climatological representativity. On approximately half

of the days, precipitation is caused by an upper level ridge with either weakly- or strongly-amplified waves. The "Previous Precipitation Event" classification is found on approximately one quarter of the days. Both "Blocking" and "Southerly Flow" setups caused precipitation only once each in the measurement period. For the model precipitation events (Fig. 10b) the fractions of the "Previous Precipitation Event" and the "Shallow Ridge" situations are smaller than for the observed events whereas the fractions of "Blocking"' and "Southerly Flow" are larger than for the measured events.

### 6.4 Moisture source regions

In Fig. 11, the horizontal and vertical course of the trajectories for the precipitation events are shown for the three arrival levels, $600\,\mathrm{hPa}$, $500\,\mathrm{hPa}$, and $300\,\mathrm{hPa}$. Ideally, the arrival level should be around the lifting condensation level determined by radiosonde data. Unfortunately, no upper-air measurements are available for the study period. Thus the calculations were done for the mentioned standard levels. The trajectories with $600\,\mathrm{hPa}$ arrival level were found not to be representative for the

25 general flow because the surface pressure exceeds $600\,\mathrm{hPa}$ for $33\,\%$ of the time in the considered period. During the rest of the time, they are strongly influenced by the near surface and thus often stay above the continent for the entire 5 days of the calculation. For the $300\,\mathrm{hPa}$ arrival level, the back-trajectories often end far above the sea surface. Furthermore, the absolute humidity is extremely low at this level (on average $0.025\,\mathrm{g\,kg^{-1}}$ compared to $0.21\,\mathrm{g\,kg^{-1}}$ at $500\,\mathrm{hPa}$). In contrast to that, many trajectories that arrive at $500\,\mathrm{hPa}$ reach a level close to the surface after five days. Also, of the three compared levels, the

30 $500\,\mathrm{hPa}$ level has on average the highest temperature. Therefore it is assumed that the flow around this level represents the main moisture transport. Similarly, the temperature at the $500\,\mathrm{hPa}$ level was assumed to be representative of $T_c$ for isotopic calculations, which is an even stronger simplification than the choice of the main transport level.





The end point of the 5 day back-trajectory was not automatically assumed to be the moisture source area: Each trajectory was cross-checked with the synoptic analysis, and the moisture source was estimated using the combined information of trajectory and general atmospheric flow conditions. Also the height of the trajectory end point, which was not always at the ocean surface, was taken into account. Pathway and transport time were adjusted according to the available information. The source regions were defined by latitude and longitude ranges. Figure 12 shows the resulting estimated source regions. The higher the assumed condensation level is, the further the spread of the source regions is, due to the increase of wind speed with altitude.

## 6.5 Isotopic modelling

In this section, the skill of MCIM and ECHAM5-wiso in reproducing the measured isotopic composition at Dome F during precipitation events is tested. The model parameters of MCIM were tuned to the measured isotopic composition of precipitation. For this, the evaporation and deposition conditions determined from the $500\,\mathrm{hPa}$ arrival level were used. The result is shown in Fig. 13 together with the measurements from the fresh snow samples. The parameters could not be adjusted to reproduce the measurements perfectly. MCIM simulates on average not enough fractionation along the transport path. The annual cycle of $\delta^{18}O$ is reproduced correctly, but it has a smaller amplitude than the measurements and a systematic offset of on average 11.6‰. It is largest in winter (Fig. 13a). The annual cycle of deuterium excess is reproduced well by MCIM as well (Fig. 13b), but it has on average a negative offset of 7.7‰. The root mean square of the deviation from observations is 13.5‰ for d and 12.3‰ for $\delta^{18}O$. Deriving the initial isotopic composition of the vapour from the ECHAM5-wiso three-dimensional vapour isotopic composition field at the end point of the backtrajectory did not improve the agreement of model and observations.

The output of ECHAM5-wiso for Dome F is shown in Fig. 13 with green circles. While both the modelled $\delta^{18}O$ and d generally have a smaller offset from the measurements in ECHAM5-wiso than in the MCIM simulations (3.1‰ for $\delta^{18}O$ and 3.8‰ for d), the annual cycle is captured better by ECHAM5-wiso only for $\delta^{18}O$. ECHAM5-wiso shows no clear annual cycle of deuterium excess. This corresponds to a relatively high rout mean square of the deviation from the measurements for d of 10.4‰ for d wheras it is only 5.6‰ for $\delta^{18}O$. Figure 14a shows the modelled $\delta^{18}O$ plotted against the measurements. $\delta^{18}O$ from ECHAM5-wiso correlates with a correlation coefficient of $R = 0.85$ with the measurements, MCIM with 0.64. The correlation of d with the measurements is higher for MCIM ($R = 0.44$) than for ECHAM5-wiso ($R = 0.21$).

There is no evidence that the reason for the inability of ECHAM5-wiso to reproduce the annual cycle of d is due to a bias in the modelled local meteorological conditions at Dome F. The $2\,\mathrm{m}$ temperature during precipitation events is on average only $0.7\,^{\circ}\mathrm{C}$ higher than the observations and the timing of the precipitation events is reproduced reasonably well by ECHAM5-wiso.

## 6.6 Influence of the weather situation and the precipitation origin on the deuterium excess

Because the weather situation has an impact on the precipitation origin, it also influences the isotopic composition of the precipitation at Dome Fuji. To investigate this, the measurements of $\delta^{18}O$ and d during different types of weather situations are compared. Of special interest is the comparison of the measurements for amplified and shallow ridges. They represent clearly different origins of precipitation, with source regions further north for amplified ridges. The deuterium excess is supposed





to contain information about the source conditions. According to the previous studies, from the higher source temperatures associated with amplified ridges, a higher d than for a more southern source region would be expected.

Table 1 shows the meteorological and isotopic measurements averaged for days with a precipitation event resulting from a shallow ridge and from an amplified ridge. For an amplified ridge the 2 m temperature at Dome F is on average 12 °C higher

and the estimated SST at the moisture source 7.6 °C higher than for a shallow ridge. The reason for the higher 2 m temperature and thus $T_c$ for amplified ridges is the advection of warmer air masses from lower latitudes. In spite of the more northern moisture source, the $\delta^{18}O$ for an amplified ridge is on average 9.2‰ higher than for a shallow ridge because the higher $T_c$ for amplified ridges reduces the temperature difference between moisture source area and final deposition site and thus causes less fractionation along the transport path. Accordingly, the deuterium excess is not, as expected, higher for an amplified ridge,

but on average 11.5‰ lower than for the shallow ridge situation since the influence of the higher $T_c$ on d is stronger than the influence of the higher SST. This is due to the temperature dependence of the ratio between the equilibrium fractionation coefficients of $HD^{16}O$ and $H_2\,^{18}O$.

Figure 15 shows the measured d plotted against SST and RH in the estimated source regions of the precipitation events. Contrary to the literature, neither for SST nor for RH a relationship with d can be detected. The correlation coefficient R

between d and SST is only -0.13, and between d and RH it is -0.14.

## 7  Discussion and conclusion

The influence of the synoptic situation and the precipitation origin on the isotopic composition of precipitation at Dome Fuji has been investigated using a 1-year set of observations. The synoptic situations causing precipitation were analysed with help of AMPS archive data. For the measurement period, 21 synoptically-induced precipitation events were identified that caused 60 %

of the total annual precipitation whereas the remaining 40 % mainly stemmed from diamond dust. This breakdown depends on the definition of "synoptic precipitation", for which a 90 % percentile threshold was used here. Better results might be achieved by crystal analysis of the precipitation samples in order to distinguish between diamond dust, snowfall, and hoar frost, as has been done for Dome C by Schlosser et al. (2015).

Whereas precipitation in the coastal areas is related to frontal activity in the circumpolar trough, synoptic precipitation

events at Dome F were found to be mostly induced by amplified planetary waves with an upper-level ridge stretching onto the Antarctic Plateau, usually associated with a strong northerly advection, which transports moisture towards Dome F.

The synoptic analysis was combined with 5-day back-trajectory calculations in order to estimate possible moisture origins for the precipitation events. A mean moisture source cantered at approximately 55 ° S was determined. Conditions at the thus-defined moisture source areas were used as input for a simple isotopic model often used in temperature reconstructions from

ice cores, the MCIM, in order to simulate the stable isotope ratios of the precipitation samples.

Both measured $\delta^{18}O$ and deuterium excess of fresh snow samples showed a clear annual cycle with a maximum of $\delta^{18}O$ in summer and an opposite cycle with a maximum in winter for the deuterium excess. Since synoptic precipitation events are mostly accompanied by advection of relatively warm air masses, $T_c$ is significantly higher during these events than on





average. $\delta^{18}O$ usually has a local maximum during precipitation events that is associated with a local temperature maximum. A corresponding local minimum in deuterium excess is found only in the winter months from June to August. MCIM was able to represent the annual cycle of the stable isotope ratios and of deuterium excess fairly well. The amount of isotopic fractionation, however, was on average underestimated by the model. While the output of an isotopically-enhanced GCM

(ECHAM5-wiso) for Dome F was on average closer to the measured isotopic composition of the snow samples, this model could not reproduce the annual cycle of deuterium excess.

In spite of the abundance of data, MCIM could not be tuned to simulate sufficient isotopic fractionation along the moisture transport path. This issue has also appeared in previous studies on the Antarctic plateau using this model (Uemura et al., 2012; Helsen et al., 2006). MCIM is a semi-empirical model that was developed in the 1980s and includes strong simplifications like

the assumption of an isolated air parcel and a single moisture source. Apart from the simplifications of MCIM, possible reasons for this disagreement are errors in the initial water vapour isotope ratio and the location and meteorological characteristics of the moisture source region. Furthermore, while the assumption that the flow at the 500 hPa level is representative for the main moisture transport to Dome F is plausible, it is a strong simplification to use the model temperature at this level as condensation temperature. An error in the modelled temperature might add to the error in the height of the condensation level. While it would

be desirable to have radiosonde data to determine the lifting condensation level, for the measuring period no upper-air data are available for Dome F.

The correlation coefficient between the measured $\delta^{18}O$ and 2 m temperature is higher for precipitation events than for the entire measurement period (that includes diamond dust) because the transport paths are mostly more clearly defined for synoptic precipitation events than for diamond dust precipitation. Also, the surface temperature inversion is usually weakened

or removed during event-type precipitation by high wind speeds and back radiation from clouds. This decreases the difference between $T_c$ and $T_{2m}$ and could cause a more direct relation of the two temperatures and thus improve the correlation of 2 m temperature and $\delta^{18}O$.

In spite of the given uncertainties in the moisture source estimation, we conclude that the results of the present study do not support the common assumption (Stenni et al., 2001; Uemura et al., 2012) that the deuterium excess of the initial water

vapour is preserved along the transport path until the final deposition at the ice core drilling site. We do not find evidence of a relationship between moisture source conditions and deuterium excess for the analysed precipitation events at Dome Fuji.

The second significant impact of the results of the present study on ice core interpretation is that, contrary to the assumption used for decades in ice core studies, a more northern moisture source does not necessarily mean a larger temperature difference between source area and deposition site and thus precipitation that is more depleted in heavy isotopes and has a higher deu-

terium excess. The warm air advection associated with the event-type precipitation discussed in this and some previous studies increases the condensation temperature considerably, thus decreasing the temperature difference between the moisture source area and the deposition site, which means less fractionation than for colder conditions.

It is not clear to what extent the results from precipitation events in a one-year period can be transferred to longer time scales. However, the physical mechanisms at play stay the same and it is important to understand how changes in the general

atmospheric circulation (e.g. during glacial-interglacial transitions) change moisture sources and transport paths for Antarctic





precipitation and thus the stable isotope ratios measured in the ice cores. This study therefore stresses the importance of Antarctic precipitation monitoring datasets for longer time periods. They also offer the advantage of not being affected by post-deposition processes associated with snow metamorphism, therefore providing the best tool to assess the meteorological controls on snowfall isotopic composition.

The fact that ECHAM5-wiso does not correctly simulate the variability of d further challenges the use of GCMs equipped with isotopes for interpretation of d for past periods.

Recently, it became possible to measure, in addition to $\delta^{18}O$ and $\delta D$, $\delta^{17}O$. The $^{17}O_{excess}$ is then calculated from $\delta^{18}O$ and $\delta^{17}O$. This variable is believed to preserve information about RH of the source region and to be independent of SST (Landais et al., 2012). Thus, it should be possible to disentangle the information about temperature and relative humidity at the

moisture source area. However, Schoenemann et al. (2014) emphasized that, like deuterium excess, $^{17}O_{excess}$ is also highly influenced by kinetic fractionation in supersaturation conditions. Thus, it may also be sensitive to temperature, especially for low temperatures at the condensation site. For a more exact quantitative climatic interpretation of stable isotope records from ice cores, further detailed process studies with present-day data combined with modelling on larger time-scales are necessary to investigate both the atmospheric and the post-depositional influences on stable isotope ratios in Antarctic snow.

*Acknowledgements.* This study was financed by the Austrian Science Fund (FWF) under grant P24223 and by the Amadée Programm of WTZ (FR 12/2014). AMPS is supported by the US National Science Foundation, Office of Polar Programs. We appreciate the support of the University of Wisconsin-Madison Automatic Weather Station Program for the Dome F AWS data set, data display, and information, NSF grant numbers ANT-0944018 and ANT-1245663. The European Centre for Medium-Range Weather Forecasts kindly provided the archive data from ERA-Interim. We thank Dr. Lindsey Nicholson for software support.



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





**Table 1.** Average values (with standard deviations in brackets) of the estimated source latitude and observations at Dome F of 2 m temperature, the estimated SST, observed precipitation, deuterium excess and $\delta^{18}O$ in precipitation for two different types of precipitation events. N is the number of days on which the event occurs

|  | N | Lat (°) | $T_{2m}$ (°C) | SST (°C) | Prec (mm w.e.) | $\delta^{18}O$ (‰) | d (‰) |
|---|---|---|---|---|---|---|---|
| Ampified ridge | 8 | -45 (7.4) | -44.3 (10.2) | 8.0 (5.4) | 0.54 (0.24) | -53.7 (8.3) | 9.5 (10.4) |
| Shallow ridge | 10 | -57 (4.7) | -56.2 (9.4) | 0.4 (1.8) | 0.42 (0.30) | -62.9 (6.9) | 21.0 (8.2) |



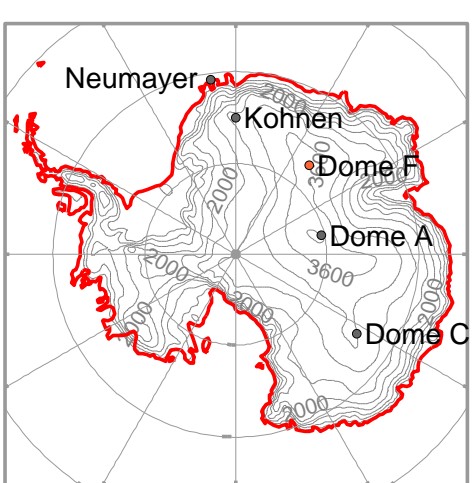

**Figure 1.** AMPS topography of Antarctica and location of selected stations



**Figure 2.** (a) Measurements and AMPS simulations of daily precipitation in mm w.e.; green dashed line: 90 % percentiles from measurements, orange dashed line: 90 % percentile from AMPS; (b) daily measurements of temperature, (c) daily measurements of $\delta^{18}O$ and (d) daily measurements of deuterium excess. The open and filled circles in (b), (c) and (d) mark the days where the precipitation exceeded the 90 % and 95 % percentile. The 90 % percentile is defined as threshold for precipitation events.




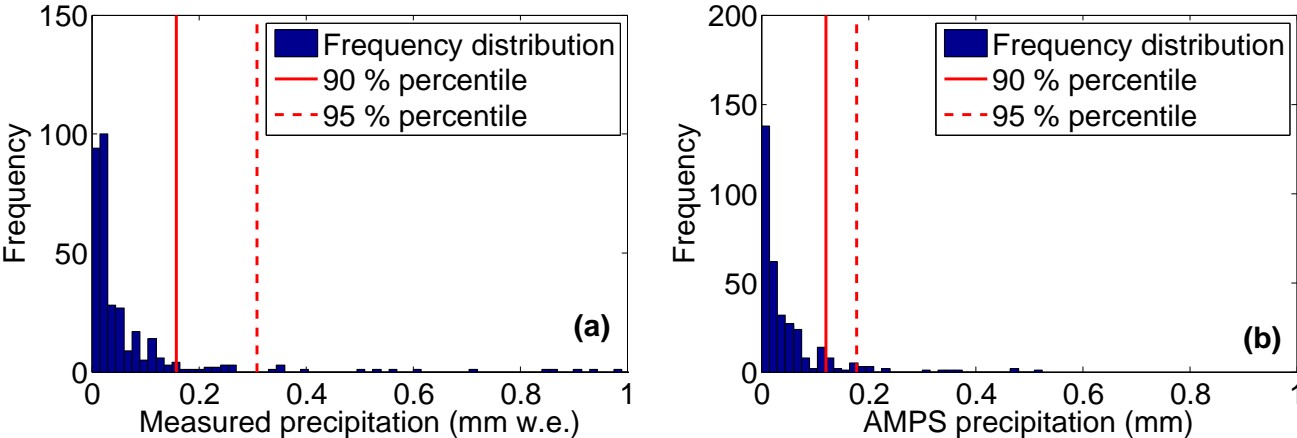

**Figure 3.** (a) Histogram of the precipitation measurements in mm w.e. with a distance of 0.015 mm w.e. between the bins; the two highest values (1.4 mm w.e. and 2.1 mm w.e) are not shown; (b) Histogram of the precipitation simulations from AMPS with the same distance of 0.015 mm between the bins

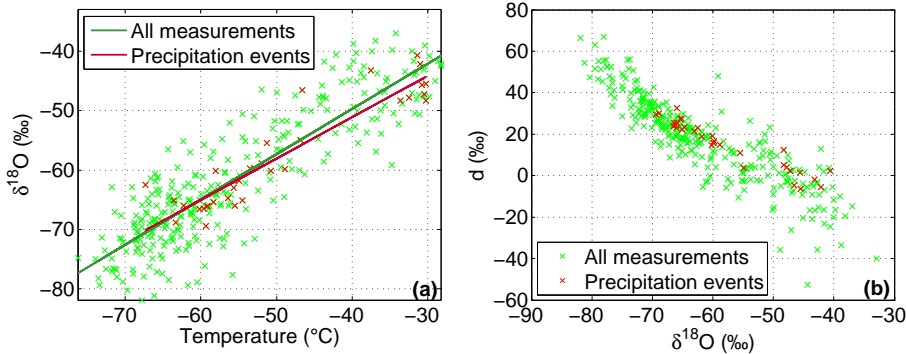

**Figure 4.** (a) Daily measurements of $\delta^{18}O$ plotted against 2 m temperature; the measurements during precipitation events are marked by red crosses. Linear fits for all measurements and for the days with a measured precipitation event are represented by a green and a read line. (b) Daily measurements of deuterium excess plotted against $\delta^{18}O$





**Figure 5.** Example for an Amplified High Pressure Ridge; (a) 500 hPa geopotential height from AMPS, 1 August 2003 12 UTC, (b) 6 h precipitation in mm from AMPS, 1 August 2003 06 UTC

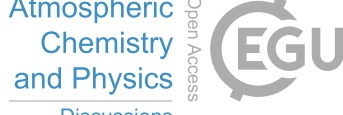

**Figure 6.** Example for a Weak High Pressure Ridge; (a) 500 hPa geopotential height from AMPS, 13 August 2003 12 UTC, (b) 6 h precipitation in mm from AMPS,14 August 2003 00 UTC





(a)                                    (b)

**Figure 7.** Example for a Blocking High; (a) 500 hPa geopotential height from AMPS, 3 November 2003 00 UTC, (b) 6 h precipitation in mm from AMPS, 3 November 2003 00 UTC





(a)                                    (b)

**Figure 8.** Example for the event Southerly Flow; (a) 500 hPa geopotential height from AMPS, 1 October 2003 00 UTC, (b) 6 h precipitation in mm from AMPS, 1 October 2003 00 UTC



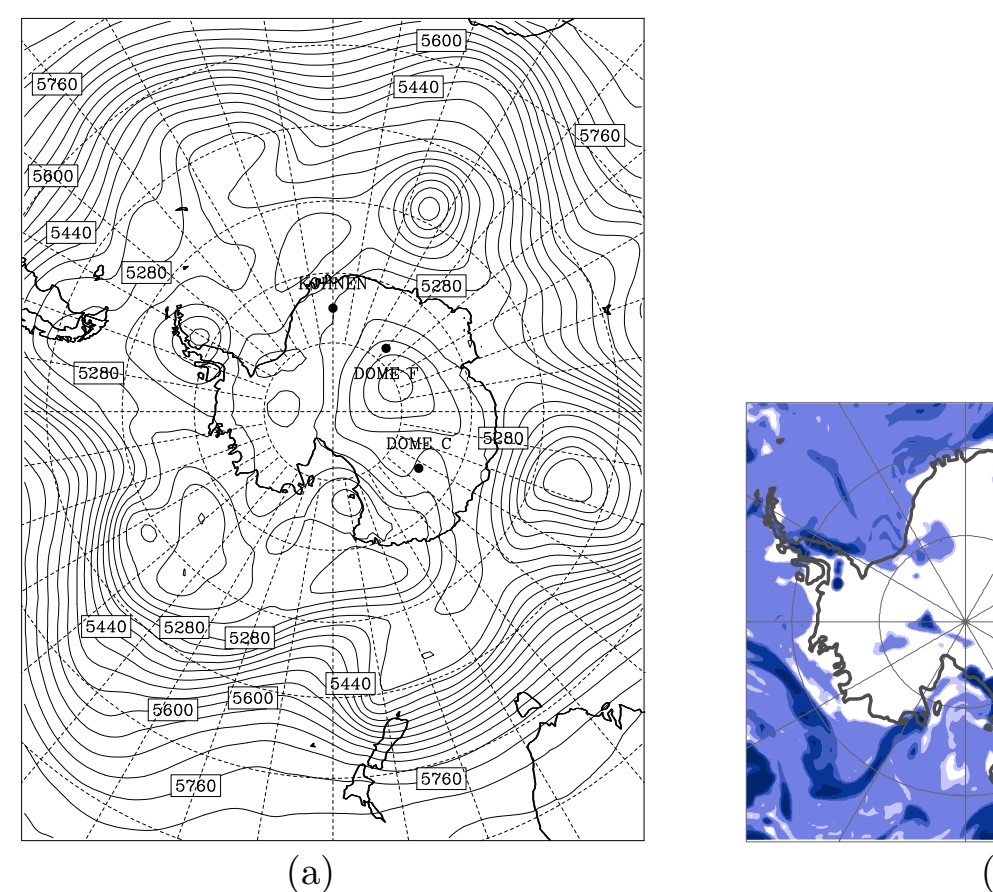

**Figure 9.** Example for the event type Previous Precipitation Event; (a) 500 hPa geopotential height from AMPS, 14 December 2003 00 UTC (b) 6 h precipitation in mm from AMPS, 14 December 2003 00 UTC

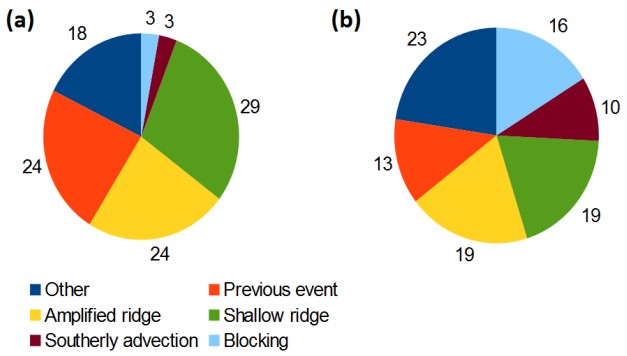

**Figure 10.** Frequency distribution of the synoptic patterns for days with precipitation events in %, for (a) the measured precipitation events and (b) the precipitation events from AMPS





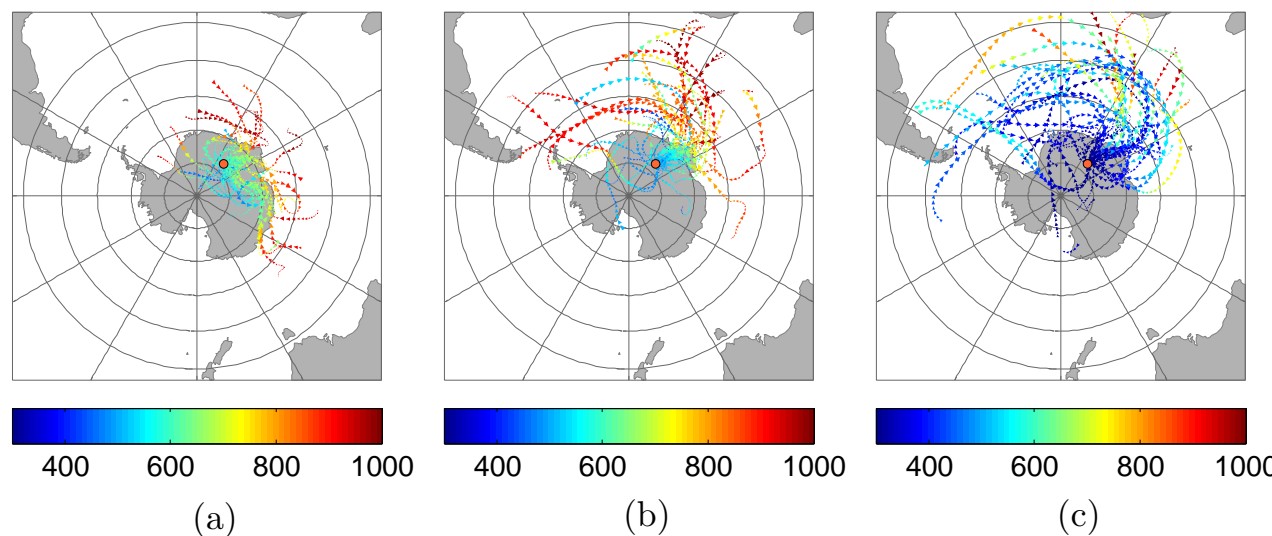

**Figure 11.** Horizontal course of the available trajectories for days with either measured or modelled precipitation events with arrival times at 00 and 12 UTC arriving at (a) 600 hPa, (b) 500 hPa and (c) 300 hPa. The colours show the pressure level in hPa. The red dot marks the location of Dome F





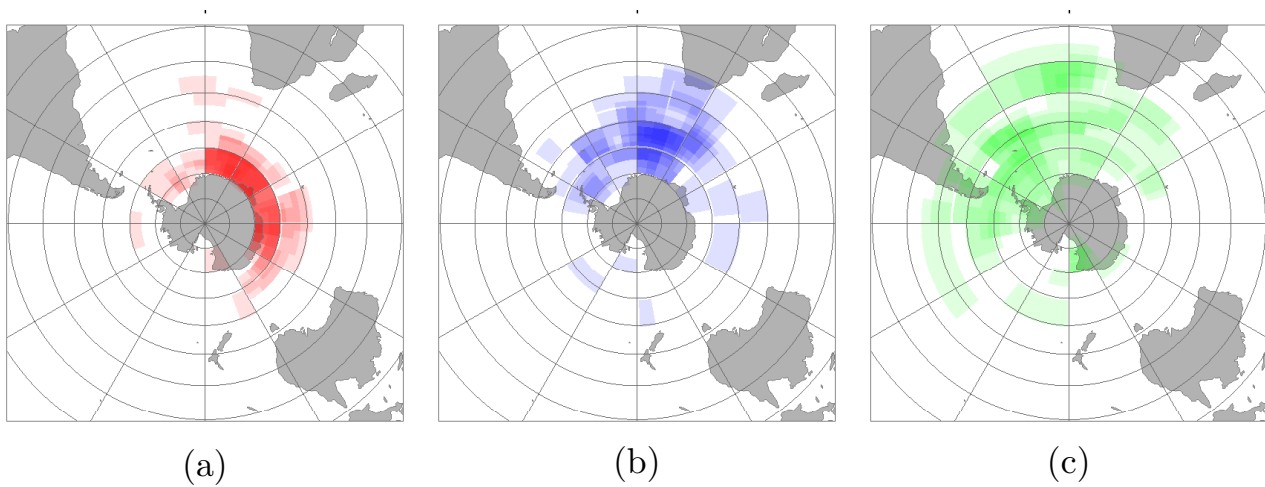

**Figure 12.** Estimated source regions of the modelled and measured precipitation events for the available trajectories arriving at (a) 600 hPa, (b) 500 hPa and (c) 300 hPa. The stronger the colour, the more frequently the respective source area was found





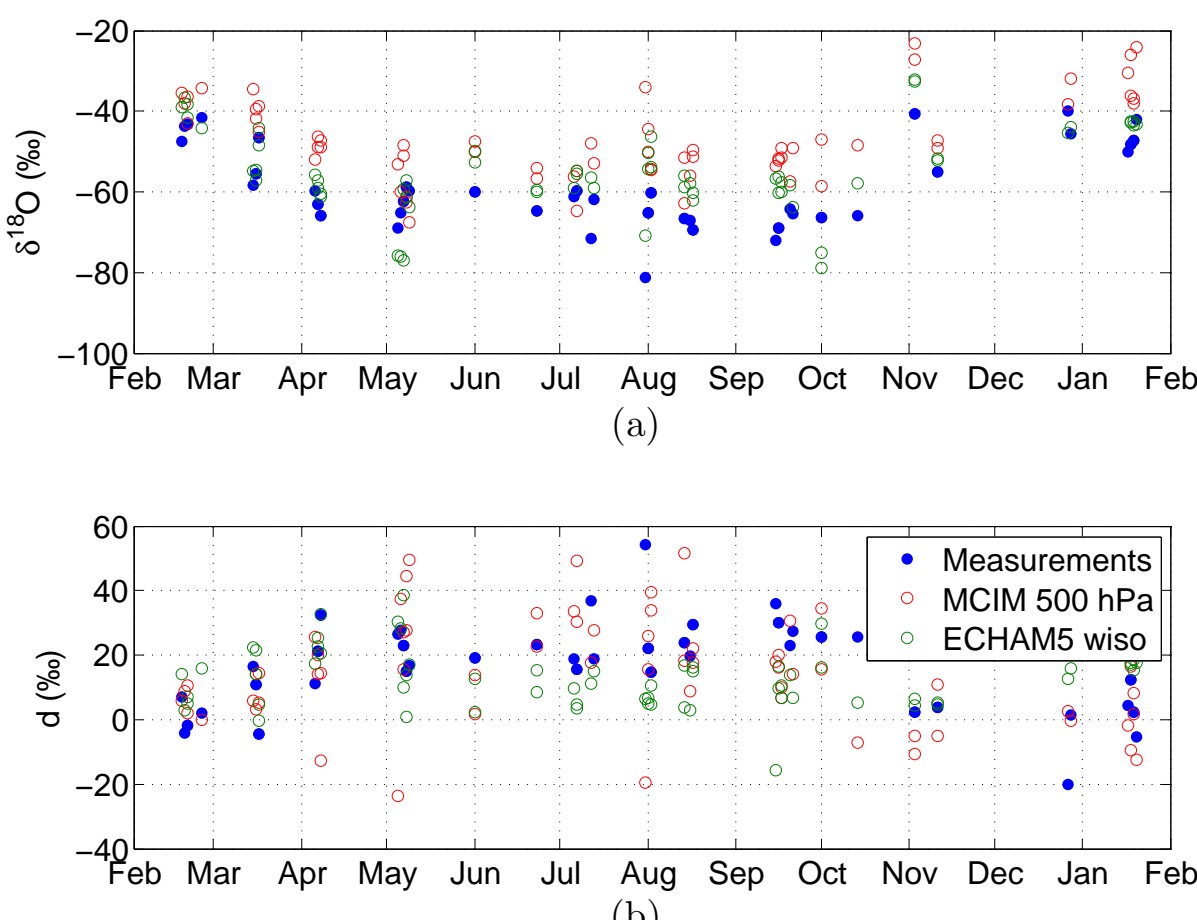

**Figure 13.** Isotopic composition of precipitation for the days with a measured or modelled precipitation event (excluding days were no trajectories were available) used from measurements, MCIM (with arrival level at 500 hPa) and ECHAM5-wiso; (a) $\delta^{18}O$ and (b) Deuterium excess




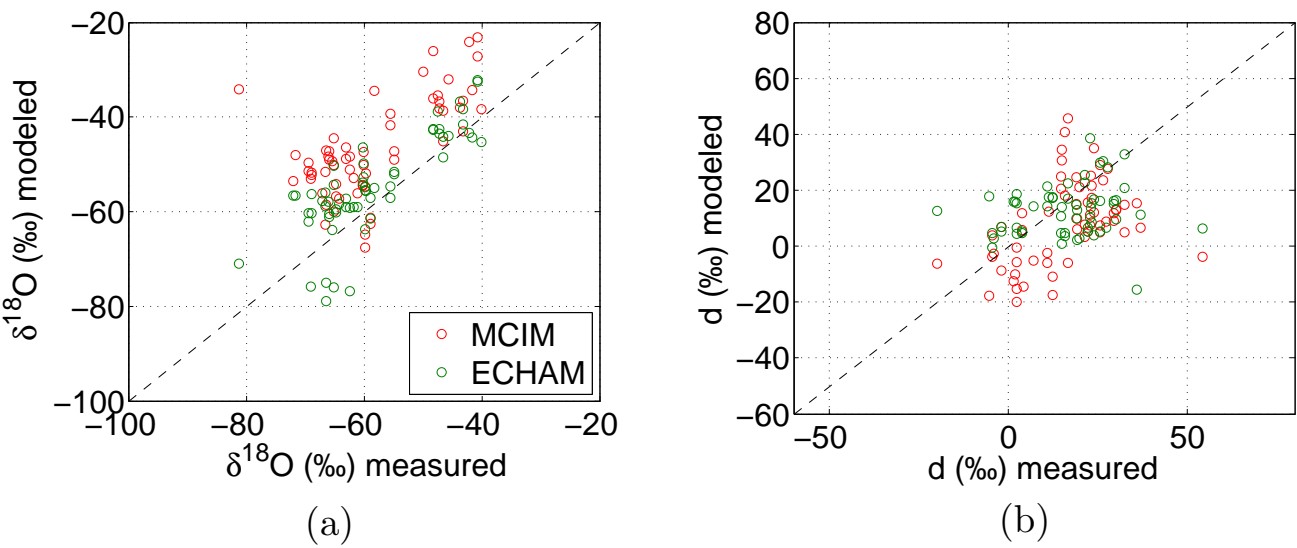

**Figure 14.** Isotopic composition of precipitation from MCIM with arrival level at 500 hPa (red circles) and ECHAM5-wiso for the same days (green circles) plotted against the measurements. The black line marks the line of equal values; (a) $\delta^{18}O$ and (b) deuterium excess

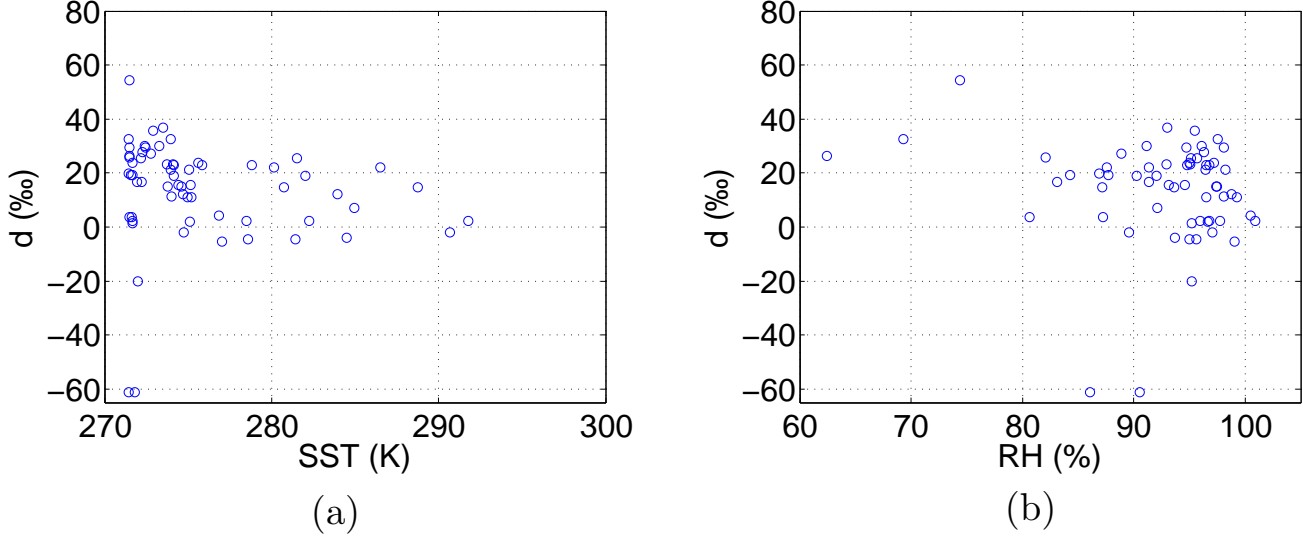

**Figure 15.** Measured deuterium excess plotted against areal mean of (a) SST and (b) RH calculated at SST from ERA-interim in the estimated source regions for days with trajectories available