# Peer review of "Precipitation regime and stable isotopes at Dome Fuji, East Antarctica"

_Atmospheric Chemistry and Physics, 2015_

## Referee Comment (RC1) · F. Parrenin (Referee) · 24 Feb 2016

This manuscript studies the regimes of precipitations and their stable isotopic composition at Dome Fuji (East Antarctica) based on 1-year observations and on modeling (AMPS, ECHAM5-wiso and MCIM).

It is found that 60% of the precipitations were caused by synoptically-induced events, while the remaining 40% are due to diamond dust (although these numbers depend on the definition used for synoptic precipitation and diamond dust).

The synoptic situations were analyzed and classified in 5 categories, with the most common being an upper-level ridge that extends onto the Antarctic plateau and causes strong northerly advection from the ocean.

[Figure]

A mean source of precipitation centered at ∼55°S was determined.

MCIM was able to reproduce the seasonal cycle of deuterium, O-18 and deuterium excess, but the isotopic fractionation was on average underestimated, even after tuning the model. ECHAM5-wiso was on average closer to the observations, but it could not reproduce the seasonal cycle of excess. This is problematic for using this kind of GCMs equipped with isotopes to interpret the deuterium excess records.

This study has consequences for ice core interpretation. It is indeed found that the relationship between d18O and surface temperature is higher for precipitations events that for diamond dust. This is an important conclusion regarding the use of the isotopic paleo-thermometer.

It is also found that deuterium excess does not have any clear link, neither with relative humidity, nor with sea surface temperature at the moisture source. This challenges the use of excess to reconstruct source conditions.

Before giving my comments, I should state that I am not an expert of atmospheric processes. So I am not well aware of the recent bibliography regarding atmospheric processes in Antarctica, although the current study gives a good overview of previous works in its section 2. That being said, I found this manuscript particularly easy to access for a non-specialist. Everything is very clear and accessible.

As an ice core scientist, I found that this study has important implications regarding the use of isotopic composition of ice to reconstruct past temperature and snow accumulation conditions at the site of deposition. For example, the correlation between snow isotopic composition and surface temperature is weak for diamond dust, which represent almost half of the precipitations at Dome Fuji. Also, the corrections based on deuterium excess applied to reconstruct past temperature variations seem to be not appropriate. Overall, I found that this study is almost ready for publication.

Minor comments :

- p. 2, l. 11-12: "In the light of .. fluctuations (Masson-Delmotte et al., 2006)" I would place this sentence right at the beginning of the introduction.

- p. 2, l. 27-28: "Noone et al. (1999)" -> "(Noone et al., 1999)"

- p. 5: Please better explain what is "Rayleigh-type model", "trajectory model", "simple isotopic models".

- p. 6, l. 28: dot after "2009)".

- p. 8, l. 1-3: what is the chosen value for n?

- p. 12, l. 21-23: I did not understand why the arrival level of precipitations is not determined by the model.

- p. 13, l. 21: "rout" -> "root"

- p. 14, l. 28: "cantered" -> "centered"

- p. 15, l. 27-28: "contrary to the assumption used for decades in ice core studies..." I am not sure this was really the assumption made in ice core studies. Ice core scientists simply said that the isotopic composition of precipitation is linked to the site minus source temperature difference, and not to the site temperature alone.

---

## Referee Comment (RC2) · Anonymous Referee #2 · 18 Mar 2016

The paper by Dittmann et al. is presenting the precipitation conditions and their influence on the water stable isotopes taking advantage of a 1-year data set of collected precipitation at the inland Antarctic site of Dome F where two deep ice cores have been retrieved in the past. The authors analysed the synoptic situation causing precipitation at Dome F using the AMPS model. The results thus obtained coupled with a back trajectory study allow them to constrain the moisture source regions. The obtained results suggest a more southerly origin for the precipitation than previously reported. Moreover, at least for the considered year, no relationship is found between deuterium excess and moisture source SST and relative humidity.

These two main conclusions point to the importance of long-term monitoring of precipitation in Antarctica in order to achieve a better interpretation of the meteorological factors affecting the variability of snowfall isotopic composition. A better understanding

of present-day processes will also, hopefully, improve the climate interpretation of the isotopic records obtained from deep ice cores. The paper is interesting, well presented and accurate and I have found the reading very smooth. I recommend its publication after minor revisions listed below.

There is one point that could be questionable if considering or not the end point of the 5 days trajectories as moisture source regions (although the authors at page 13 are claiming that the end point of the 5 days was not automatically assumed as moisture source area). The approach of Sodemann and Stohl (2009) was pointing to a more northern origin for moisture sources and as such how the relationships between deuterium excess and SST/h should be considered?

Page 2, line 5: delete "ice".

Page 2, line 19: the deuterium excess should be defined here adding also the citation (Dansgaard, 1964).

Page 2, line 20: add also wind speed in parenthesis.

Page 2, 27-28: change into (Noone et al., 1999).

Page 6, line 2: I would change the sentence "Dome F is ….. in Anatrctica." into something like "Dome F is one of the places where very old ice can be found".

Page 6, line 3: may you add the depth of the first ice core drilling?

Page 6, line 5: may you add that the EPICA Dome C is covering the past 800,000 years (Jouzel et al., 2007)?

Page 6, line 8: the sentence that Fujita and Abe were the first to perform direct precipitation measurements is not completely true: there have been also other two cases: one is Ekaykin et al. (2004) presenting 1-year precipitation data from Vostok and then a quite old paper by Aldaz e Deutsch (1967) (Aldaz L. & Deutsch S., 1967. On a relationship between air temperature and oxygen isotope ratio of snow and firn in the

South Pole region. Earth Planet. Sci. Lett., 3, 267-274).

Page 6, line 14: I am not completely convinced that sublimation processes could be ruled out especially in summer and probably explaining the negative values of deuterium excess.

Page 6, line 28: add a dot after ". . ..2009)".

Page 8, line 27-28: please, check here the English.

Page 11, line 17: change from August to November.

Page 13, line 1: add a dot after ". . . source area".

Page 14, line 28: correct "cantered" into centred.

Page 22, figure 2: the orange dash line is not clear at all. May you improve? In the caption: bottom line: add respectively after percentile.

Page 23, figure 4 caption: correct "read" into red.

---

## Author Response (AR1)

The original text of the comment is black, the answers are blue.

This manuscript studies the regimes of precipitations and their stable isotopic composition at Dome Fuji (East Antarctica) based on 1-year observations and on modeling (AMPS, ECHAM5-wiso and MCIM).

It is found that 60% of the precipitations were caused by synoptically-induced events, while the remaining 40% are due to diamond dust (although these numbers depend on the definition used for synoptic precipitation and diamond dust).

The synoptic situations were analyzed and classified in 5 categories, with the most common being an upper-level ridge that extends onto the Antarctic plateau and causes strong northerly advection from the ocean.

A mean source of precipitation centered at ∼55◦S was determined.

MCIM was able to reproduce the seasonal cycle of deuterium, O-18 and deuterium excess, but the isotopic fractionation was on average underestimated, even after tuning the model. ECHAM5-wiso was on average closer to the observations, but it could not reproduce the seasonal cycle of excess. This is problematic for using this kind of GCMs equipped with isotopes to interpret the deuterium excess records.

This study has consequences for ice core interpretation. It is indeed found that the relationship between d18O and surface temperature is higher for precipitations events that for diamond dust. This is an important conclusion regarding the use of the isotopic paleo-thermometer.

It is also found that deuterium excess does not have any clear link, neither with relative humidity, nor with sea surface temperature at the moisture source. This challenges the use of excess to reconstruct source conditions.

Before giving my comments, I should state that I am not an expert of atmospheric processes. So I am not well aware of the recent bibliography regarding atmospheric processes in Antarctica, although the current study gives a good overview of previous works in its section 2. That being said, I found this manuscript particularly easy to access for a non-specialist. Everything is very clear and accessible.

As an ice core scientist, I found that this study has important implications regarding the use of isotopic composition of ice to reconstruct past temperature and snow accumulation conditions at the site of deposition. For example, the correlation between snow isotopic composition and surface temperature is weak for diamond dust, which represent almost half of the precipitations at Dome Fuji. Also, the corrections based on deuterium excess applied to reconstruct past temperature variations seem to be not appropriate. Overall, I found that this study is almost ready for publication.

We thank the referee for his positive review. We address each comment below.

Minor comments :

- p. 2, l. 11-12: "In the light of .. fluctuations (Masson-Delmotte et al., 2006)" I would place this sentence right at the beginning of the introduction.

We changed this in the revised manuscript.

- p. 2, l. 27-28: "Noone et al. (1999)" -> "(Noone et al., 1999)"

We corrected this in the revised manuscript.

- p. 5: Please better explain what is "Rayleigh-type model", "trajectory model", "simple isotopic models".

Rayleigh-type models belong to the simple isotopic models. The explanation was improved in the text (p.5,l15):

To reconstruct information about the temperature both at the deposition site and at the source region, simple Rayleigh–type models are used. These models consider the fractionation in an isolated air parcel using only moisture source and condensation conditions as input. They do not include dynamic processes or turbulent mixing. In a pure Rayleigh model it is assumed that the condensate in

the cloud is immediately removed by precipitation after formation (e.g. Merlivat and Jouzel, 1979). In other simple models (e.g. Ciais and Jouzel, 1994), the fraction of the condensate that is precipitated can be chosen.

p.8,l.10 was shortened to:
Generally, two types of isotopic models are distinguished: Rayleigh-type models (see above) and isotopic GCMs (e.g. ECHAM5-wiso, Werner et al., 2011), which include an explicit representation of stable water isotopes into a three dimensional atmospheric model.

p. 5 l. 21: To explain the term "trajectory model", a sentence was added:
"Trajectory models use three dimensional atmospheric fields to trace an air parcel back from the precipitation site to the evaporation area."

- p. 6, l. 28: dot after "2009)".
We corrected this in the revised manuscript.
- p. 8, l. 1-3: what is the chosen value for n?
For simplicity's sake, RIP does not define a threshold for convergence, but simply does two iterations for each time step, which turned out to be exact enough in the praxis for our purposes. The time step we used was 600s.
- p. 12, l. 21-23: I did not understand why the arrival level of precipitations is not determined by the model.
AMPS archive data were used to determine the best approximation of the mean arrival level. In order to determine the exact arrival level for each case the lifting condensation level has to be determined, for which the model data are not accurate enough because of the low humidity of the atmosphere above the Antarctic plateau.
- p. 13, l. 21: "rout" -> "root"
We corrected this in the revised manuscript.
- p. 14, l. 28: "cantered" -> "centred"
We corrected this in the revised manuscript.
- p. 15, l. 27-28: "contrary to the assumption used for decades in ice core studies..." I am not sure this was really the assumption made in ice core studies. Ice core scientists simply said that the isotopic composition of precipitation is linked to the site minus source temperature difference, and not to the site temperature alone.
The isotopic composition of precipitation is mainly linked to the temperature difference between source area and deposition site. However, quite a few studies consider not only this difference, but also the geographical location of the moisture source. For the site temperature, the warm air advection described in our study is usually not taken into account in those studies.
We changed our formulation as follows and gave an additional reference:
Contrary to the assumptions used for many years in ice core studies (see Jouzel, 2014 and references therein)….

**Referee #2**

The original text of the comment is black, the answers are blue.

The paper by Dittmann et al. is presenting the precipitation conditions and their influence on the water stable isotopes taking advantage of a 1-year data set of collected precipitation at the inland Antarctic site of Dome F where two deep ice cores have been retrieved in the past. The authors analysed the synoptic situation causing precipitation at Dome F using the AMPS model. The results thus obtained coupled with a back trajectory study allow them to constrain the moisture source regions. The obtained results suggest a more southerly origin for the precipitation than previously reported. Moreover, at least for the considered year, no relationship is found between deuterium excess and moisture source SST and relative humidity.

These two main conclusions point to the importance of long-term monitoring of precipitation in Antarctica in order to achieve a better interpretation of the meteorological factors affecting the variability of snowfall isotopic composition. A better understanding of present-day processes will also, hopefully, improve the climate interpretation of the isotopic records obtained from deep ice cores. The paper is interesting, well presented and accurate and I have found the reading very smooth. I recommend its publication after minor revisions listed below.

We thank the referee for the helpful comments and the positive review.

There is one point that could be questionable if considering or not the end point of the 5 days trajectories as moisture source regions (although the authors at page 13 are claiming that the end point of the 5 days was not automatically assumed as moisture source area). The approach of Sodemann and Stohl (2009) was pointing to a more northern origin for moisture sources and as such how the relationships between deuterium excess and SST/h should be considered?

We stress in the paper that we do not automatically assume the end point of the 5-day trajectories to be the moisture sourc area. Also, we cannot exactly DETERMINE the moisture sourece, it is just an estimate. However, it is possible to clearly distinguish between a moisture source in the polar ocean and one at lower latitudes. We do not think that trajectories over longer time spans as used in the approach of Sodemann and Stohl (2009) would change the result that no relationship between deuterium excess and SST/RH can be found. In the comparison of the average d for two weather situations with clearly different source regions (Table 1) no signal of the source latitude in the deuterium excess of snow could be found. Even if the exact position of the source regions might be questionable (see above), the synoptic difference between the two situations (amplified ridge and shallow ridge) clearly points to a more northern source region for the amplified ridge. SST mostly depends on the latitude. RH, however, shows stronger spatial and temporal fluctuations and a weaker dependence on latitude. Thus the uncertainty of the source RH and its relationship with deuterium excess is certainly higher. We agree that more detailed studies over a longer time span, including measurements of water vapour isotope ratios, are necessary to further investigate the relationship between deuterium excess and the source conditions.

Page 2, line 5: delete "ice".

We changed this to "snow (and thus ice)" in the revised manuscript. It is true that in the mentioned studies snow was analysed but the same mechanism are relevant for ice.

Page 2, line 19: the deuterium excess should be defined here adding also the citation (Dansgaard, 1964).

After "deuterium excess" we added the definition : (d = δD−8·δ 18 O, Dansgaard (1964))) and deleted the definition on p.4 l. 24.

Page 2, line 20: add also wind speed in parenthesis.

We added "wind speed" in the revised manuscript.

Page 2, 27-28: change into (Noone et al., 1999).

We corrected this in the revised manuscript.

Page 6, line 2: I would change the sentence "Dome F is . . ... in Anatrctica." into something like "Dome F is one of the places where very old ice can be found".

We changed the sentence into: Dome F is one of the places in Antarctica where very old ice can be found.

Page 6, line 3: may you add the depth of the first ice core drilling?

We added the depth (2503 m).

Page 6, line 5: may you add that the EPICA Dome C is covering the past 800,000 years (Jouzel et al., 2007)?

We added this in the revised manuscript.

Page 6, line 8: the sentence that Fujita and Abe were the first to perform direct precipitation measurements is not completely true: there have been also other two cases:
one is Ekaykin et al. (2004) presenting 1-year precipitation data from Vostok and then a quite old paper by Aldaz e Deutsch (1967) (Aldaz L. & Deutsch S., 1967. On a relationship between air temperature and oxygen isotope ratio of snow and firn in the South Pole region. Earth Planet. Sci. Lett., 3, 267-274).

To our knowledge, Ekaykin et al. 2004 did not perform direct precipitation measurements but analysed data from snow pits. With direct precipitation measurements we mean sampling and determining the amount of fresh snow immediately after the snowfall.

Thanks a lot for the advice. We are usually careful not to overlook the work of the early researchers. We mentioned the work of Aldaz and Deutsch and added the reference in the revised version.

We added the following sentences on P.4, L.4:

Few studies have performed direct precipitation measurements on the Antarctic Plateau. Aldaz and Deutsch (1967) sampled fresh snow for isotope analysis already in 1964/65 at the South Pole, without measuring precipitation amounts, though. Additionally, they used radiosonde data to determine the lifting condensation level to be able to relate the temperature at this level to the stable isotope ratios of precipitation. At Dome C, direct daily precipitation samples have been analysed since 2006 (Schlosser et al.,2015). The measurements include the precipitation amount, analysis of stable water isotopes and crystal structure analysis.

Furthermore, we changed the sentence: Fujita and Abe (2006) were the first to perform direct precipitation measurements and sampling for isotopic measurements
on the Antarctic Plateau. (P. 6, L. 8) to
… in Central Antarctica.

Page 6, line 14: I am not completely convinced that sublimation processes could be ruled out especially in summer and probably explaining the negative values of deuterium excess.

We agree that sublimation may play a role when the sampling took place several hours after the snowfall (and maybe even during the snowfall), but we can exclude the reoccurring cycle of sublimation and deposition that is very poorly understood by investigating fresh snow samples. We

changed the text to "….. alterations through wind scouring and sublimation after the snowfall are reduced to a minimum."

The deuterium excess is usually anti-correlated with 18O/temperature and thus negative values occur in summer, even without sublimation, the latter could only explain part of it.

Page 6, line 28: add a dot after ". . ..2009)".

We corrected this in the revised manuscript.

Page 8, line 27-28: please, check here the English.

We changed it to: In ECHAM5, additionally to $H_2O^{16}$, water containing $^{18}O$ and D has been implemented in the water cycle. For each phase change, fractionation processes are considered.

Page 11, line 17: change from August to November.

We corrected this in the revised manuscript.

Page 13, line 1: add a dot after ". . . source area".

We used the colon since the following sentence explains the "not automatically assumed".

Page 14, line 28: correct "cantered" into centred.

We corrected this in the revised manuscript.

Page 22, figure 2: the orange dash line is not clear at all. May you improve? In the caption: bottom line: add respectively after percentile.

We added "respectively".

We agree that the line did not come out well in the pdf –file. We have changed this in the revised version.

Page 23, figure 4 caption: correct "read" into red.

We corrected this in the revised manuscript.

[revised manuscript text omitted]

5    Few studies have performed direct precipitation measurements in Antarctica. Aldaz and Deutsch (1967) sampled fresh snow for isotope analysis already in 1964/65 at the South Pole, without measuring precipitation amounts, though. Additionally, they used radiosonde data to determine the lifting condensation level to be able to relate the temperature at this level to the stable isotope ratios of precipitation. At Dome C, daily precipitation samples have been analysed since 2006 (Schlosser et al., 2016). The measurements include the precipitation amount, analysis of stable water isotopes and crystal structure analysis.

10 ## 2.2 Origin of precipitation

[revised manuscript text omitted]

To reconstruct information about the temperature both at the deposition site and at the source region, simple Rayleigh-type models are used. These models consider the fractionation in an isolated air parcel using only moisture source and condensation conditions as input. They do not include dynamic processes or turbulent mixing. In a pure Rayleigh model it is assumed that the condensate in the cloud is immediately removed by precipitation after formation (e.g. Merlivat and Jouzel, 1979). In other simple models (e.g. Ciais and Jouzel, 1994), the fraction of the condensate that is precipitated can be chosen. A frequently-used model is the mixed cloud isotope model (MCIM) (Ciais and Jouzel, 1994). It calculates the isotopic fractionation of an isolated air parcel on the cooling path from the first evaporation to the final deposition. It considers equilibrium and kinetic fractionation processes. Several studies have tried to test the performance of the model to reproduce the present day isotopic composition of snow from measurements (Jouzel and Merlivat, 1984; Petit et al., 1991; Uemura et al., 2012). To include atmospheric dynamics and to find appropriate source conditions some studies have combined Rayleigh-type models with trajectory models. Trajectory models use three-dimensional atmospheric fields to trace an air parcel back from the precipitation site to the evaporation area. 
[revised manuscript text omitted]